# Unusual Quadrupedal Locomotion in Rat during Recovery from Lumbar Spinal Blockade of 5-HT_7_ Receptors

**DOI:** 10.3390/ijms22116007

**Published:** 2021-06-02

**Authors:** Urszula Sławińska, Henryk Majczyński, Anna Kwaśniewska, Krzysztof Miazga, Anna M. Cabaj, Marek Bekisz, Larry M. Jordan, Małgorzata Zawadzka

**Affiliations:** 1Nencki Institute of Experimental Biology, Polish Academy of Sciences, 02-093 Warsaw, Poland; h.majczynski@nencki.edu.pl (H.M.); a.kwasniewska@nencki.edu.pl (A.K.); k.miazga@nencki.edu.pl (K.M.); a.cabaj@nencki.gov.pl (A.M.C.); m.bekisz@nencki.edu.pl (M.B.); m.zawadzka@nencki.edu.pl (M.Z.); 2Department of Physiology and Pathophysiology, University of Manitoba, Winnipeg, MB R3E 0J9, Canada; Larry.Jordan@umanitoba.ca

**Keywords:** quadrupedal locomotion, 5-HT receptors, lumbar spinal cord

## Abstract

Coordination of four-limb movements during quadrupedal locomotion is controlled by supraspinal monoaminergic descending pathways, among which serotoninergic ones play a crucial role. Here we investigated the locomotor pattern during recovery from blockade of 5-HT_7_ or 5-HT_2A_ receptors after intrathecal application of SB269970 or cyproheptadine in adult rats with chronic intrathecal cannula implanted in the lumbar spinal cord. The interlimb coordination was investigated based on electromyographic activity recorded from selected fore- and hindlimb muscles during rat locomotion on a treadmill. In the time of recovery after hindlimb transient paralysis, we noticed a presence of an unusual pattern of quadrupedal locomotion characterized by a doubling of forelimb stepping in relation to unaffected hindlimb stepping (2FL-1HL) after blockade of 5-HT_7_ receptors but not after blockade of 5-HT_2A_ receptors. The 2FL-1HL pattern, although transient, was observed as a stable form of fore-hindlimb coupling during quadrupedal locomotion. We suggest that modulation of the 5-HT_7_ receptors on interneurons located in lamina VII with ascending projections to the forelimb spinal network can be responsible for the 2FL-1HL locomotor pattern. In support, our immunohistochemical analysis of the lumbar spinal cord demonstrated the presence of the 5-HT_7_ immunoreactive cells in the lamina VII, which were rarely 5-HT_2A_ immunoreactive.

## 1. Introduction

Quadrupedal locomotion is an essential animal behavior based on the rhythmic coordinated movement of four limbs that is regulated by the neuronal structures located within the central nervous system, in the brain, and spinal cord. The movement of each limb is controlled by its own spinal locomotor neuronal network called Central Pattern Generator (CPG), located in cervical segments for left and right forelimbs and in upper lumbar segments for left and right hindlimbs. The CPGs responsible for the four limb movements during quadrupedal locomotion control a basic motor activity pattern of muscles belonging to the antagonistic flexor-extensor groups and are modulated by peripheral somatosensory feedback [1,2,3,4]. Coordination of left-right limb movements of the fore girdle (cervical) or hind girdle (lumbar) is mediated by commissural interneurons [5,6,7,8,9,10,11,12,13]. Although considerable data are available on the left-right hindlimb coordination mechanisms, relatively less is known about the control of the forelimb-hindlimb coupling [14,15,16,17]. Communication between CPGs of fore- and hindlimbs is achieved by ascending and descending propriospinal pathways projecting ipsilaterally and diagonally [13,17,18,19,20,21,22,23,24,25]. However, it remains unknown whether serotonin controls the function of the inter girdle propriospinal connections.

The activity of the motor spinal cord structures is controlled by supraspinal systems, among which the serotonergic (5-hydroxytryptamine; 5-HT) descending pathway from the brainstem is critical. Data from the isolated neonatal rat spinal cord preparation show that serotonin receptor agonists are sufficient to activate the spinal CPGs for locomotion and that this effect can be blocked by 5-HT antagonists [26,27]. Subsequent work showed that 5-HT_2A_ and 5-HT_7_ receptors are important for this action of 5-HT [28,29,30,31,32,33]. Despite many pieces of evidence for the role of 5-HT in locomotion coming from the neonatal preparations or spinal cord injury experiments, the mechanisms of 5-HT action in controlling locomotion in adult animals is not clear. Our recent data demonstrated that blockade of the 5-HT_2A_ receptors by intrathecal application of cyproheptadine at the lumbar spinal cord induced significant alteration of interlimb hindlimb coordination as well as reduction of locomotor electromyographic (EMG) amplitude that was followed by total paralysis [33]. We have also shown that blockade of the 5-HT_7_ receptors by intrathecal application of SB269970 in adult intact rats suppressed locomotion by partial paralysis of hindlimbs with disrupted intra- and inter-hindlimb coordination [32]. Here, we investigated the role of the 5-HT_7_ and 5-HT_2A_ receptors in the control of limb coordination in quadrupedal locomotion in adult freely moving rats, which remained unknown so far.

Previously, we have reported the effects of blockage of the 5-HT_7_ and 5-HT_2A_ receptors on hindlimb locomotor movements when focusing the attention on the changes obtained before total paralysis in the last rhythmic EMG activity associated with rather limited hindlimb movements (a short time after intrathecal drug application, usually <5 min) [32,33]. Here, we analyzed the four-limb locomotor patterns in the time of the first sign of recovery of quadrupedal locomotion (ELRP; the early locomotor recovery period) after hindlimb paralysis induced by intrathecal application of SB269970 or cyproheptadine (antagonists of 5-HT_7_ and 5-HT_2A_ receptors, respectively). Investigation of the pattern of the quadrupedal locomotion was carried out using chronic EMG activity recorded from selected forelimb (FL) and hindlimb (HL) muscles in the freely moving rats on the treadmill, which allowed us to establish the ipsilateral and diagonal interlimb coordination between fore- and hind girdles as well as the left-right interlimb coordination within each girdle. We identified an unusual pattern in the quadrupedal locomotion with a double stepping rhythm in forelimbs in relation to hindlimb stepping (2FL-1HL) during locomotor recovery after blockage of 5-HT_7_ receptors by SB269970 application, in contrast to the unaffected locomotor pattern observed during recovery from blockade of 5-HT_2A_ receptors by cyproheptadine application. Moreover, we verified the distribution of the 5-HT_7_ and 5-HT_2A_ immunoreactive cells in the rat lumbar spinal cord segments and found the presence of the 5-HT_7_ immunoreactive cells in the lamina VII, which rather rarely was 5-HT_2A_ immunoreactive. Our functional and morphological results allowed us to conclude that the 5-HT_7_ but not 5-HT_2A_ receptors on lumbar interneurons contribute to the spinal modulation of quadrupedal locomotion in adult intact rats. We suggest that the lumbar interneurons located in lamina VII and controlled by 5-HT_7_ receptors, with ascending projections to the forelimb CPG can be responsible for the observed 2FL-1HL effect.

## 2. Results

In our previous papers [32,33], we described that intrathecal application of SB269970 (an antagonist of 5-HT_7_ receptors) or cyproheptadine (an antagonist of 5-HT_2A_ receptors) at the lumbar spinal cord level of Wistar adult rats induced in a short time (<5 min) a transient paralysis of both hindlimbs, which lasted for several minutes. In the time of paralysis, the animals moved forward using their forelimbs, when the hindlimbs lost their postural and locomotor function. Here we investigated the later effect of blockade of defined 5-HT receptors on the locomotor performance on the treadmill at the rats’ preferred belt speed ranging between 10 to 20 cm/s at the time when a first clear recovery of four-limb locomotion appeared (the early locomotor recovery period; ELRP). The recovery of quadrupedal locomotion was observed 21 min (median) after SB269970 and 40 min (median) after cyproheptadine application. Careful visual inspection of locomotor performance allowed us to discover that the recovered four-limb locomotor behavior after blockage of 5-HT_7_ receptors was accompanied by the different patterns of stepping in forelimbs than in hindlimbs. We have found that blockage of 5-HT_7_ receptors in the lumbar spinal cord of adult rats facilitated at the time of ELRP a doubled rhythm in the forelimbs in relation to the hindlimb stepping (2FL-1HL; 2 forelimb steps-1 hindlimb step). In contrast, when 5-HT_2A_ receptors were blocked by cyproheptadine application in the six rats (out of eight tested with SB269970) the four-limb locomotor pattern remained unaffected: the locomotor pattern in forelimbs remained coupled to that of hindlimbs (1FL-1HL). These results were consistent regardless of the cannula tip locations, which were identified in the various lumbar spinal cord levels from L1 to L6 (L1/L2 in 3 rats, L3/L4 in 2 rats L5/L6 in 3 rats) in investigated animals.

The EMG activity of fore- and hindlimb muscles recorded in this experimental condition allowed us to find the frequent appearance of a double EMG burst in the Tri muscles in relation to a single EMG burst cycle of the ipsilateral Sol or TA muscles during locomotion on a treadmill after blockage of 5-HT_7_ receptors (Figure 1A–D) but not after blockage of 5-HT_2A_ receptors (Figure 1E–H). 

The effect of SB269970 was present in eight rats, while the effect of cyproheptadine treatment was tested in six out of these eight rats. The linear envelopes of rectified and integrated EMG activity demonstrate that the forelimb cycles of the Tri EMG burst activity remained locked with the hindlimb cycles in a way of the 2FL-1HL coupling after SB269970 application (Figure 1D) while in pre-drug (Figure 1C,G) and after cyproheptadine (Figure 1H) the regular 1FL-1HL pattern was observed in the EMG burst activity of the left and right leg muscles (this will be investigated further using circular analysis). Although in the analyzed episodes of locomotor activity the fragments of the 2FL-1HL patterns were stable and consistent, occasionally, they could be interspersed with single stepping of the normal 1FL-1HL pattern (see example in Figure 1B).

### 2.1. Interlimb Coordination in Pairs of Limbs of Fore- and Hind Girdles

First, we investigated the correctness of the locomotor pattern in fore- and hindlimb using the circular analysis to establish left-right interlimb coordination in both girdles separately during quadrupedal locomotion on the treadmill, which recovered in the time of ELRP after SB269970 (8 rats) and after cyproheptadine (six rats out of eight tested with SB269970). Our analysis (Figure 2A,B) demonstrated the presence of a very consistent alternate pattern in both girdles close to 180° phase shift between onsets of EMG bursts of left-right homologous muscles (Tri muscles in the forelimbs and TA muscles in the hindlimbs) that was associated with high strength of interlimb coordination (expressed as *r*-value > 0.75; *p* < 0.001, Rayleigh’s circular statistical tests). The circular analysis demonstrated clear interlimb left-right coordination in fore- as well as hindlimb movements, which was at a similar level both before and after either drug application (Figure 2A; Table 1). 

SB269970 did not change the interlimb coordination strength (*r*-values) both between left and right Tri as well as TA muscles (Kruskal-Wallis test for four groups of data shown in Figure 2A; KW(3) = 3.94, *p* = 0.27). SB269970 also did not affect phase shift between these left and right homologous muscles (*p* ≥ 0.73; Watson-Williams angular tests). Note that this clear left-right coordination of limb in both girdles was obtained after SB269970 application despite the presence of the 2FL-1HL coupling pattern in the quadrupedal locomotion on a treadmill (see following analysis). Similar to SB269970, the above parameters were also unchanged after the administration of cyproheptadine (Figure 2B; Table 1). The Kruskal-Wallis test did not show any significant differences between mean *r*-values (Figure 2B; KW(3)= 2.95, *p* = 0.4). Finally, cyproheptadine did not affect the mean phase shift calculated between left and right Tri (*p* = 0.61; Watson-Williams test) and TA muscles (*p* = 0.039; Watson-Williams test; but this *p*-value did not remain significant after 5% FDR correction for multiple comparisons). Individual mean values of the described above parameters and the appropriate pre-drug vs. post-drug comparisons *p*-values are shown in Table 1.

### 2.2. Fore-Hind Limb Coupling

Next, we analyzed the fore-hindlimb coupling during quadrupedal locomotion by establishing the Regularity Index (RI), which we defined as the ratio of the number of steps performed by the forelimb (number of cycles according to the triceps EMG burst activity) to the number of steps performed by the hindlimb (number of cycles according to the soleus EMG burst activity) during spontaneous locomotion on a treadmill (Figure 3). In control pre-drug quadrupedal locomotion RI was approximately equal to 1 in both experimental conditions (the values calculated before injection of SB269970 for the left and right muscles were respectively 1.01 ± 0.05 and 0.99 ± 0.02 and analogous values obtained before cyproheptadine use: 1.07 ± 0.10 and 1.00 ± 0.05; for each one-sample Wilcoxon test for differences from 1, the *p*-value was ≥0.25). This indicates that for one step taken by the forelimb there is exactly one step taken by the hindlimb. After SB269970 application an impairment of fore-hindlimb coupling was observed. The rhythmic double stepping of forelimbs occurred often along with normal hindlimb stepping, causing the RI value to increase. This was observed during the ELRP (median 21 min of post-drug time) when the first sign of four-limb locomotor recovery occurred after hindlimb paralysis induced by SB269970 application. Thus, the mean RI values for left and right fore-hindlimb coupling were around 1.5 (respectively 1.48 ± 0.22 and 1.52 ± 0.20; *p* = 0.008, for both one-sample Wilcoxon tests). In other words, the increased RI after the blockade of 5-HT_7_ receptors by SB269970 indicates, on average, a 50% bigger number of steps in forelimbs in relation to the number of steps performed by hindlimbs (Figure 3A). It should be noted that the episodes of locomotor performances with the 2FL-1HL forelimb-hindlimb coupling after the blockade of 5-HT_7_ receptors were sometimes interspersed with the walking periods of the normal 1FL-1HL pattern, which could be in different rats differently expressed (see an example of EMG recording in Figure 1B). As a consequence of the changes which SB269970 induced to the forelimb stepping pattern, RI coefficients were significantly higher than those obtained for the control situation (Kruskal-Wallis test performed for four groups of data (left and right control, as well as left and right post-drug RI factors) KW(3) = 24.04; *p* < 0.0001; multiple comparisons *p* = 0.0016 for pre-drug vs. post-drug RI comparisons for left legs and *p* = 0.0002 for right limbs).

Contrary to SB269970, a typical 1FL-1HL stepping pattern was observed during quadrupedal locomotion obtained during the ELRP after cyproheptadine application (at median 40 min of post-drug time), where RI values obtained both for left and right fore-hindlimb coupling remained unchanged compared to the control situation (Figure 3B) and remained very close to 1 (respectively 1.04 ± 0.07 and 1.01 ± 0.02; *p* ≥ 0.25 for both one-sample Wilcoxon tests). After cyproheptadine application RI remained unchanged (KW(3) = 2.77; *p* = 0.43).

### 2.3. Step Cycle Duration Analysis

In this part of the analysis, we investigated whether the 2FL-1HL pattern was associated with different step cycle durations in the fore- and hindlimbs. The cycle duration was established as a time difference between the onset of two consecutive EMG bursts of the Tri muscle in FLs while in HLs the TA muscle EMG bursts were used. We found that after SB269970 application, the cycle duration in FL was significantly shorter when compared to the pre-drug conditions, which was reflected as a reduced ratio of post-drug to the pre-drug ones (0.73 ± 0.15 for left FL and 0.73 ± 0.14 for right FL; one-sample Wilcoxon tests, *p* = 0.016 for both comparisons that remained significant after 5% FDR correction for multiple comparisons; Figure 4A). In contrast to FLs, for HLs SB269970 did not induce any changes in cycle duration and the ratio post- to pre-drug remained close to 1 (1.09 ± 0.24 for left HL and 1.11 ± 0.29 for right HL; one-sample Wilcoxon tests, *p* = 0.64 and *p* = 0.46 respectively). As a consequence of selective shortening of the FL step cycle, the ratio became smaller than those obtained for HLs (Kruskal-Wallis test for four groups of data, KW(3) = 16.57, *p* = 0.0009; with following post-hoc test *p =* 0.0034 for left limbs and *p* = 0.0047 for right limbs; both remained significant after 5% FDR correction for multiple comparisons; Figure 4A).

After cyproheptadine application the cycle duration in FLs as well as in HLs remained unchanged compared to the pre-drug condition and the ratio to the pre-drug value remained close to 1 (1.06 ± 0.20 for left FL, 1.02 ± 0.19 for right FL, 0.98 ± 0.16 for left HL, 1.00 ± 0.17 for right HL; one-sample Wilcoxon test, *p* ≥ 0.99 for each comparison; Figure 4B). After cyproheptadine the relative values of cycle duration also did not differ between fore- and hindlimbs (Kruskal-Wallis for 4 groups of data, KW(3) = 0.33; *p* = 0.95).

Summarizing, we would like to mention that shortening of the forelimb cycle duration after the blockage of 5-HT_7_ receptors is rather understandable due to the appearance of the 2FL-1HL pattern of locomotion, but the results presented here additionally indicated that hindlimb cycle duration remains in the same time unchanged compared to control pre-drug values.

### 2.4. Amplitude of EMG Activity

The effect of the application of both drugs on the EMG burst amplitude in the FL and HL muscles during quadrupedal locomotion in the time of ELRP was investigated in the next step (Figure 5). Similar to the cycle duration, we analyzed the EMG burst amplitude after normalization to the appropriate pre-drug value. In principle, during ELRP after intrathecal injection of both 5-HT_7_ and 5-HT_2A_ receptor antagonists we did not notice any changes in the burst amplitude of hindlimb Sol as well as TA muscles. Although, after SB269970, we observed a small increase in the Sol burst amplitude of the right HL (1.21 ± 0.23 times larger than the control one; one-sample Wilcoxon test, *p* = 0.039) and a trend for such an effect for the Sol muscles of the left HL (1.22 ± 0.26 times; one-sample Wilcoxon test, *p* = 0.055; Figure 5A), these *p*-values did not remain significant after 5% FDR correction for multiple comparisons. Furthermore, there was no change in the Sol EMG burst amplitude after cyproheptadine (1.09 ± 0.29—for the left HL and 1.08 ± 0.27—for the right HL; one-sample Wilcoxon test, *p* = 0.44 for both comparisons) and both drugs applications (SB269970 and cyproheptadine) did not cause any change in the burst amplitude of TA muscles (0.91 ± 0.33 and 1.04 ± 0.27, respectively for the left and right HL after SB269970 application; 0.82 ± 0.21 and 0.89 ± 0.13, respectively for the left and right HL after cyproheptadine application; one-sample Wilcoxon test, *p* ≥ 0.16 for each comparison; Figure 5B). There was also no difference between the effects of either drug on the EMG burst amplitude of Sol (Kruskal-Wallis test, KW(3) = 0.79, *p* = 0.85) as well as TA muscles (KW(3) = 2.09, *p* = 0.55). Since the Sol and TA EMG amplitude remained similar to the appropriate pre-drug value and did not differ depending on the type of the drug used in the experiment, these results indicate that at this time of recovery from hindlimb paralysis there is no visible effect of both drugs on the Sol and TA motoneuron pool activity.

In contrast to data obtained for the HL muscles, there was a twofold increase in the Tri EMG burst activity in both left and right FL muscles (Figure 5C) after cyproheptadine (1.95 ± 0.56 and 2.37 ± 0.68 for left and right muscle respectively; one-sample Wilcoxon test, *p* = 0.031 for both comparisons) and 1.5-fold increase in the right FL after SB269970 (1.56 ± 0.61; one-sample Wilcoxon test, *p* = 0.039). All these three *p*-values remained significant after 5% FDR corrections for multiple comparisons. The increase in the EMG burst amplitude of the left Tri muscle, despite being quite large (1.38 ± 0.48), was statistically not significant (one-sample Wilcoxon test, *p* = 0.15). Moreover, Kruskal-Wallis analysis showed some differences within all four groups of amplitude data for Tri muscles (KW(3) = 8.8, *p* = 0.032) and indicated more specifically that the amplitude increment after cyproheptadine was more pronounced than after SB269970 in the right Tri EMG burst (appropriate post-hoc test, *p* = 0.018 and this value remained significant after 5% FDR correction). However, cyproheptadine did not induce a larger amplitude increase than SB269970 in the case of the activity of the left Tri muscle (post-hoc test, *p* = 0.13).

The increased amplitude in the Tri EMG burst activity after SB269970 can be explained by shorter step cycles in forelimbs, which are typically associated with faster locomotor patterns [34]. However, the Tri EMG amplitude increment after cyproheptadine, which was even greater than the burst amplitude increase of the right Tri EMG burst after SB269970 application, was not related to the shorter cycle duration with increased speed of forelimb movement. The alternative explanation might be that the blockade of 5-HT_2A_ receptors on elements of lumbar networks, which send their projections up to the cervical network (CPGs and MNs), affects forelimb motoneuron activity. Thus, we next investigated whether this is related to some alteration of coordination between limbs of fore- and hind-girdles.

### 2.5. Intergirdle Ipsi- and Diagonal Coordination

Next, to determine whether the coordination between ipsilateral and diagonal pairs of limbs during a locomotor performance on the treadmill at the time of ELRP is affected by blockade of 5-HT_7_ or 5-HT_2A_ receptors, the circular analysis was performed. To investigate the coordination of the pair of ipsilateral limbs (i.e., the phase shift as well as the coordination strength), the onsets of the EMG burst activity recorded from the Tri muscle in the forelimbs versus the hindlimb cycle based on the Sol muscle EMG burst activity was analyzed. Circular analysis of the interlimb coordination between ipsilateral fore- and hindlimb extensor muscles (Tri vs. Sol) demonstrated that the limbs moved approximately in the opposite phase in both pre-drug conditions (Figure 6A,C; Table 2 and Table 3) and this did not change after cyproheptadine (Figure 6C; Table 3). 

In contrast to cyproheptadine, during quadrupedal locomotion after SB269970 application (at the time of ELRP) ipsilateral phasing was significantly altered presenting the 2FL-1HL pattern. It is important to emphasize that in the case of the 2FL-1HL pattern significant coordination between fore and hindlimb stepping could only be obtained when the first and second populations of Tri EMG bursts were analyzed separately. This is how the analyzes were performed, the results of which are presented below. It appeared that after SB269970 the ipsilateral phase values for the first and second forelimb EMG bursts clustered around two values on polar plots of around 110–115° and 220–225°, i.e. approximately 18–19% of a cycle earlier and 11–13% of a cycle later than the corresponding phase cluster obtained for the control pre-drug condition (Figure 6A; Table 2). For left muscles, the comparison with mean phase shift measured during the control period before SB269970 administration gave a very significant difference in the case of 1st post-drug cluster (*p* < 0.001, Watson-Williams test) and show some trend for the difference in the case of 2nd one (*p* = 0.069, Watson-Williams test). Similar analysis of right muscles revealed that average phase shift values obtained both for the 1st and 2nd post-drug clusters were highly different from the control phase shift measured between EMG bursts before SB269970 application (*p* ≥ 0.003, Watson-Williams tests).

Similar to the results of phase shift analysis, cyproheptadine application also did not cause any clear differences in the coordination strength of the ipsilateral pair of limbs. The corresponding mean *r*-values were quite similar before and after application of this antagonist both for left and right pairs of muscles (Kruskal-Wallis test for all 4 groups of data shown in Figure 6C; KW(3) = 1.35, *p* = 0.72; compare lengths of blue mean vectors on Figure 6C and mean *r*-values included in Table 3). Interestingly, despite substantial changes in ipsilateral phase relationships and the appearance of the 2FL-1HL pattern, SB269970 caused only a relatively small difference in the coordination strength (Figure 6A; Table 2). Kruskal-Wallis statistical analysis carried out on *r*-values between all six data groups (i.e., on one control group and two post-drug groups—for the 1st and 2nd EMG burst in Tri activity—and each of them for both left and right side of the animal body), showed an overall significant result (KW(5) =11.79; *p* = 0.04). However, for most of the post-hoc comparisons the differences between mean *r*-values were insignificant (Table 2). This was the case when comparing left-side coordination between pre- and post-drug conditions (*p* ≥ 0.063) and also when comparing the right-side coordination in the case of the 1st Tri burst (*p* = 0.034, but this *p*-value did not remain significant after 5% FDR correction for multiple comparisons). In fact, in the case of SB269970 only right-side coordination *r*-values measured for the 2nd Tri EMG burst differed obviously between pre- and post-drug conditions (*p* = 0.008, and this *p*-value remained significant after 5% FDR correction), where the coordination after SB269970 was significantly weaker (Figure 6A; Table 2).

To investigate the coordination of the diagonal pairs of limbs, the onsets of individual EMG bursts of the FL Tri muscle were measured in relation to the contralateral HL cycle established on the base of the corresponding TA EMG bursts. It appeared that, similarly to the results of the analysis of the ipsilateral coordination, cyproheptadine did not affect diagonal coordination between FLs and HLs (Figure 6D; Table 3). On the other hand, after the blockade of 5-HT_7_ receptors by SB269970 application, as for ipsilateral coordination, the presence of two clusters of diagonal phase shifts (separate for the 1st and 2nd Tri EMG burst) was identified. In the case of diagonal coordination, the mean phase shift established during the control pre-drug condition was around 70°. Not different value was obtained during ELRP after application of SB269970 for phases of the 1st Tri EMG burst (*p* ≥ 0.12, Watson-Williams tests), while phases of the 2nd burst were on average about 140°–150° greater than control-ones (*p* < 0.0001, Watson-Williams tests; Figure 6B and Table 2). In conclusion, we can state that in the case of the diagonal coordination obtained during the 2FL-1HL pattern, occurring during ELRP after intrathecal SB269970 application, the 1st bursts in the “double” Tri EMG activity appeared in similar moments of the reference cycle of TA as the single Tri muscle bursts obtained in the control situation (approximately in the middle of the 1st quarter of the reference cycle of TA muscle). On the other hand, the 2nd Tri EMG bursts in the 2FL-1HL pattern appeared much later, approximately in the middle of the 3rd quarter of the reference cycle of TA muscle.

Regarding the strength of diagonal coordination (*r*-value), cyproheptadine did not cause any clear changes (Kruskal-Wallis test for four groups of *r*-value data in Figure 6D; KW(3) = 2.26, *p* = 0.52; compare also lengths of blue mean vectors on Figure 6D and mean *r*-values in Table 3). Moreover, similar to the ipsilateral coordination, SB269970 induced much less noticeable differences in diagonal coordination strength than those in phase relations. As for ipsilateral coordination, Kruskal-Wallis statistical analysis carried out on *r*-values between all 6 data groups (i.e., on one control group and two post-drug groups—for the 1st and 2nd EMG burst in Tri activity—and each of them for both left and right animal side), showed an overall weakly significant result (KW(5) =11.59; *p* = 0.04). However, for most of the post-hoc comparisons, the differences were insignificant (Figure 6B and Table 2). Similar mean *r*-values were obtained for pre- and post-drug analysis of the right Tri—left TA coordination both for the 1st and 2nd Tri burst (*p* ≥ 0.088) and also when comparing left Tri—right TA coordination in the case of the 2nd Tri burst (*p* = 0.085). The clear difference between pre- and post-drug conditions was present only for left Tri—right TA coordination *r*-values measured for the 1st Tri muscle EMG burst (*p* = 0.006, and this *p*-value remained significant after 5% FDR correction), where the coordination after SB269970 was significantly weaker (Figure 6B; Table 2).

However, we would like to mention one additional fact regarding all coordinations established after the SB269970 application. Although significantly lower *r*-vector values were obtained during the post-drug condition in two individual coordinations, all 10 *r*-vector values for interlimb coordination in each girdle as well as for ipsilateral and diagonal coordination between girdles established after application of 5-HT_7_ receptor antagonist were decreased compared to pre-drug conditions. It is very unlikely that such a situation can occur accidentally (*p* = 0.0016, sign test). Therefore, it seems that the mean strength of the whole group of 10 measured coordinations was diminished (by around 14%; pre-drug and post-drug group average *r*-values were respectively: 0.86 ± 0.02 and 0.74 ± 0.04). So, summarizing the whole coordination analysis in the case of SB269970, we can state that unusual locomotor pattern of double forelimb stepping occurring during ELRP after blockade of 5-HT_7_ receptors remained significantly locked with the hindlimb cycles in a way of the 2FL-1HL coupling in the ipsi- as well as diagonal coupling. However, coordination between girdles was less accurate and more variable when the rats were adjusting interlimb coordination to maintain dynamic stability during the 2FL-1HL locomotor performance.

### 2.6. 5-HT_7_ and 5-HT_2A_ Expression in Lumbar Segments of Adult Spinal Cord

Finally, we analyzed the distribution of 5-HT_2A_ and 5-HT_7_ receptors in the rat lumbar spinal cord. An analysis of immunohistochemical staining showed that the 5-HT_7_ receptor immunoreactive cells were distributed evenly across the lumbar segments (Figure 7A). In addition to the dorsal horn positive staining, the 5-HT_7_ positive small cells were present in laminae VII and VIII (Figure 7A,B). In the ventral horn, some small 5-HT_7_ positive cells were identified in lamina IX among bigger cells of motoneuron phenotype (Figure 7A,C), which occasionally also were immunoreactive, although this was not evident in all lumbar sections. The 5-HT_2A_ receptor immunoreactivity was also observed across the lumbar segments, however, the pattern of staining was different from that of 5-HT_7_. In addition to the dorsal horn positive staining, the 5-HT_2A_ positive staining was particularly associated with big cells located in the ventral horn lamina IX (motoneuron area, Figure 7D,E). In contrast to 5-HT_7_, relatively fewer 5-HT_2A_ immunoreactive cell bodies were present in the lamina VII (Figure 7D,F). When comparing the immunoreactivity of both types of receptors in the lumbar spinal cord segments, two differences should be emphasized, the small cells in the lamina VII (propriospinal interneurons area) were mainly 5-HT_7_ but not 5-HT_2A_ positive, but the bigger cells of motoneuron phenotype in lamina IX were 5-HT_2A_ positive but very seldom 5-HT_7_ positive. Overall, the staining pattern of the 5-HT_2A_ and 5-HT_7_ receptor immunoreactivity was similar for all the lumbar sections. 

Considering our functional results we can conclude that the blockade of 5-HT_7_ receptors on propriospinal interneurons of lamina VII together with limited effect on motoneurons might be responsible for different functional effects that we observed after the SB269970 but not after cyproheptadine, which application was affecting mainly motoneurons in lamina IX but seldom interneurons in lamina VII.

## 3. Discussion

Our results show that affecting 5-HT_7_ receptors in the lumbar spinal segments by SB269970 intrathecal application can result in modification of the fore-hind limb locomotor pattern during rat quadrupedal locomotion on the treadmill. This effect was reflected by the presence of the stable 2FL-1HL forelimb-hindlimb coupling locomotor pattern in the time of ELRP after blockade of 5-HT_7_ but not after blockade of 5-HT_2A_ receptors by cyproheptadine. Our results indicate that 5-HT_7_ but not 5-HT_2A_ receptors located on cells in the lumbar spinal cord regulate the coordination of fore-hind limb locomotion in adult rats in the post-drug time of the ELRP. During these episodes of quadrupedal locomotion, the forelimb cycles remain locked with the hindlimb cycles in the pattern of the 2FL-1HL stepping. At the same time, the stepping of left-right limbs in the fore- and hind-girdles analyzed separately remained significantly coordinated in the symmetrical alternating way. We conclude that 5-HT_7_ receptors located on long ascending propriospinal interneurons in the lamina VII of the lumbar spinal segments sending axons to the cervical spinal cord segments might be responsible for the control of coordination of the fore- and hindlimb CPG networks during quadrupedal treadmill locomotion in adult rats.

### 3.1. The Different Rhythms in Fore- and Hindlimbs in Locomotion of Intact Mammals

It has been demonstrated that adult intact cats can step faster with forelimbs than hindlimbs on a transverse split-belt treadmill when taking two steps by forelimbs within one hindlimb step cycle [35,36]. The analysis showed that although bilateral coordination in the girdles was less accurate and more variable, the animals were able to adjust interlimb coordination and maintain dynamic stability during such locomotor performance when the forelimb belt was moving faster than the hindlimb belt. As in our data, they also found that during these episodes of the 2FL-1HL walking pattern the forelimb cycles remain locked with the hindlimb cycles [36]. The fact that intact cats can produce the 2FL-1HL coupling when the forelimbs step faster than the hindlimbs during quadrupedal locomotion on a transverse split-belt treadmill suggests that it must be a natural stable locomotor pattern of the normal mammalian spinal cord neural circuitry supported by the functional interaction between the fore- and hindlimb neural networks depending on task requirements [24,35,36]. The 2FL-1HL forelimb-hindlimb coupling in the time of ELRP after the blockade of 5-HT_7_ receptors by intrathecal application of SB269970 expresses a specific type of fore-hindlimb coordination that might indicate the emergence of a new stable state of the four-limb CPG network activity. The mechanisms that might be responsible for such interlimb coupling with a 2FL-1HL forelimb-hindlimb pattern are poorly understood. The main functional connections between networks controlling limbs of the fore and hind girdles are supported by propriospinal interneurons that interconnect spinal segments by descending or ascending axons [23,24,37,38]. The cell bodies of the propriospinal interneurons were identified within cervical/upper thoracic (descending projection) as well as lumbar (ascending projection) enlargements in lamina VII and VIII [37]. It is known that the long propriospinal neurons that originate in the lumbar enlargement and terminate in the cervical enlargement ascend in the ipsilateral VLF simultaneously sending collaterals to the contralateral side of the spinal cord [18,21,39,40,41]. As the cervical CPG network activity can be driven by the lumbar CPG [19], long ascending propriospinal axons are the major candidates to be involved in coupling the lumbar and cervical CPGs. Our results indicate that the mechanisms responsible for the 2FL-1HL pattern might be related to the 5-HT_7_ receptors localized on the elements of the neural network in the lumbar spinal cord segments that send ascending projection to the cervical enlargement to modulate the forelimb CPG networks.

### 3.2. The Different Rhythms in Fore and Hindlimb in Locomotion of Mammals with Partial Spinal Cord Lesions

A 2FL-1HL locomotor pattern can be observed after partial spinal lesions in rats [42,43] and cats [17,44]. Górska with coworkers [42] suggested that a faster forelimb rhythm in rats with partial spinal cord lesions could be a result of a compensatory strategy developed due to the reduced inhibitory influence from hindlimb CPGs to forelimb CPGs. Our group also reported such 2FL-1HL fore-hindlimb coupling in the hemisected rats in 5–7 days after spinal cord injury in the locomotor recovery process (Figure 1B; [45]). The mechanism of this walking strategy that is manifested as a 2FL-1HL coupling remains unclear. Thibaudier et al. [17] proposed that a 2FL-1HL coupling is a way to maximize animal stability during walking. Moreover, the shorter forelimb steps could be a strategy to maintain the center of gravity within the support plane obtained between the different points of each animal limb contacting the ground during quadrupedal locomotion [42,46]. Here we observed a 2FL-1HL pattern during locomotor activity after blockade of 5-HT_7_ receptors. Our analysis shows that the EMG amplitude of Sol and TA muscles after the blockade of 5-HT_7_ receptors is not different from that of the pre-drug conditions. In contrast to that, the Tri EMG amplitude demonstrates slightly increased amplitude. This difference in amplitude of the Tri EMG and adjusted interlimb 2FL-1HL coordination may reflect a necessity for maintaining dynamic stability during quadrupedal locomotion controlled by the affected neural network by 5-HT_7_ receptor modulation. Similar mechanisms for the 2FL-1HL locomotor pattern were proposed in cats walking on the split-belt treadmill [36]. We suggest that for this effect, ascending propriospinal connections from lumbar segments with 5-HT_7_ receptors supporting communication between the forelimb and hindlimb CPGs might be responsible. However, because such a 2FL-1HL pattern can appear also after partial spinal cord injury there might be some other pathways involved in the modulation of the fore-hindlimb relationship. Further investigations are needed to explore this issue.

### 3.3. Candidates of Genetically Identified Propriospinal Connections for Control of Fore and Hindlimb Locomotor Pattern

Quadrupedal locomotion in mammals depends on the recruitment of defined populations of spinal neurons. Recent investigations using transgenic mice allowed researchers to define specific populations of such neurons, based on their specific transcription factor expression, and describe their functional roles for locomotor performance [47,48,49,50,51]. The involvement of interneurons located in the ventral progenitor domains (V0–V3) and the dorsal interneuron 6 (dI6) in the locomotion received particular attention in many laboratories around the world (e.g., [48,52]).

There is considerable evidence from mouse transgenic models about the role of V1 (En1 positive) and V2b (Gata3 positive) inhibitory interneurons in the regulation of extensor-flexor motor rhythm [23,53,54] and about the role of V0-Dbx1 and V2a-Chx10 in left-right limb coordination related to short ascending or descending projections [55,56,57]. In contrast to that, less is known about the genetic identity of long propriospinal interneurons projecting between the cervical and lumbar networks controlling quadrupedal locomotion. Kiehn with his group investigated the role of the V0-Dbx1 neurons in quadrupedal locomotion in mice. They found that mice lacking V0-Dbx1 neurons were not able to present alternating limb movements, instead, the only gait they expressed was hopping [50,51,58]. Deletion of inhibitory V0_D_ (Pax7-derived) interneurons impaired left-right alternation at slow to medium speeds but not at fast speeds. While ablation of excitatory V0_V_ (Pax7-negative) cells led to the opposite pattern; left-right alternation was present at slow speeds but was lost at fast speeds. This group of researchers focused their attention mainly on the commissural connections and did not discuss the role of long propriospinal connections responsible for fore-hind limb coordination, which was also affected in their investigations i.e., when changing from symmetrical quadrupedal gait to bound. Ruder with coworkers [22] investigated the spinal networks controlling forelimbs and hindlimbs and described that both CPG networks (cervical and lumbar) are bidirectionally connected by mono- and polysynaptic neural pathways. They identified excitatory and inhibitory long propriospinal interneurons descending from cervical to lumbar networks, while the ascending long propriospinal connections were in the vast majority excitatory. The lumbar neurons targeting cervical spinal levels were found to belong to two progenitor domains; V0-Dbx1 neurons were targeting contralaterally, while V2-Shox2 neurons were targeting ipsilaterally. Ruder with coworkers [22] found that the selective ablation of descending spinal neurons connecting cervical and lumbar networks impaired quadrupedal locomotion by reducing speed during exploration and by perturbing interlimb coordination during fast locomotion on a treadmill. They found that V0v, but not V0_D_ neurons contribute to excitatory contralateral long descending projections, while V2-Shox2 neurons contribute to ipsilateral descending projection. However, they did not investigate the effect of ablation of lumbar interneurons with ascending projections on locomotor performance. Recently Pocratsky and co-workers [25] provided interesting data suggesting that in rats long ascending interneurons may form a flexible task-specific network for securing fore-hind limb coordination in a context-dependent manner. Conditional silencing of the long ascending interneurons resulted in various alterations of locomotor performance from a mild disruption of left-right hindlimb coordination to the alteration of quadrupedal pattern where hindlimbs moved in synchrony while forelimbs were galloping and then to the pattern where the fore and hindlimbs moved in synchrony. It is interesting that when after silencing of the ascending interneurons a feature of the walk or trot was disrupted, still a stable locomotor pattern was maintained [25]. Given the above, the populations of the lumbar interneurons with ascending projection to the cervical spinal CPGs might be responsible for the effect of the 2FL-1HL pattern obtained after SB269970 intrathecal application in our investigations.

Recent investigations addressed a question to the role of dI6 neurons in locomotion [59,60,61]. The subsets of dI6 cells that express the transcription factors WT1 or DMRT3 are located in lamina VII and VIII of the postnatal mouse spinal cord [52,60,62]. Their potential role in the regulation of locomotor activity was proposed based on data indicating their rhythmic activity during fictive locomotion [52,60,63]. Mice lacking the *Wt1* gene display an increased number of uncoordinated steps during locomotor activity on a treadmill, which was associated with a declining number of commissural interneurons [59]. Haque with coworkers ([60] described complementary data showing that dI6-Wt1 are inhibitory neurons projecting contralaterally and their acute silencing induces left-right alterations during fictive locomotion recorded in spinal cord preparation of newborn mice. What is interesting, in horses, a non-sense mutation in the *Dmrt3* gene facilitates lateral gaits (i.e., pace) and inhibits the transition from trot to gallop, while mice lacking this gene, display problems in hindlimb movement coordination as well as major difficulties running at higher speeds [64,65]. These authors concluded that the *Dmrt3* neurons display a critical role for left-right coordination and also for coordinating fore-hind limb movements. Further work is needed to answer whether the populations of these interneurons (V0-V3 and/or dI6) are controlled by serotonin and which 5-HT receptors are involved.

### 3.4. Ascending Inter-Girdles (Lumbo-Cervical) Connections and 5-HT Innervation

There is growing evidence demonstrating the crucial role of serotonin in the control of the motor spinal cord structures responsible for quadrupedal locomotion in mammals. It is also known that lumbar neural circuitry can drive the neural cervical network, which controls forelimb muscle activity [19]. This was investigated in neonatal rats, in which activity of both CPG networks controlling fore- and hindlimb muscles was modulated by 5-HT. In adult rats, Reed et al., [40] described the columns of the neuronal population of ascending lumbo-cervical long propriospinal neurons located in laminae VII and VIII in the lumbar spinal cord segments, which control interlimb coordination during four-limb locomotion. Our results suggest that in adult rats these lumbar interneurons with long ascending projection to the cervical network might be under serotonergic control with the mechanism of 5-HT_7_ receptors.

In contrast to long ascending propriospinal interneurons connecting the lumbar with cervical CPGs, there is more information about 5-HT control of short ascending commissural interneurons. The short ascending commissural interneurons, which have axons crossing in the middle, are rhythmically activated by 5-HT in embryonic slice spinal cord preparation [66]. Moreover, mice lacking the population of the commissural interneurons V0-Dbx1 display left-right uncoupling during fictive locomotion induced by 5-HT and NMDA bath application [57]. These ascending commissural interneurons located in lamina VII and VIII in the thoracolumbar segments of the normal neonatal mice spinal cord preparation were excited by 5-HT [67]. The mechanism by which 5-HT exerts the action on these ascending propriospinal interneurons was not identified in these investigations [57,67]. Liu with coworkers [68] demonstrated the crucial role of 5-HT_7_ receptors in the control of locomotor activity in rodents. Using 5-HT_7_ receptor knockout mice they showed that 5-HT produced only uncoordinated rhythmic activity characterized by the absence of alternation between ipsilateral flexor-extensor and right-left ventral roots. In the same investigation, they showed that in neonatal spinal cord preparation of wild-type mice, 5-HT could evoke coordinated locomotor activity that was next blocked by SB269970 (5-HT_7_ antagonist). These results confirm that the 5-HT_7_ receptors are required for 5-HT induced locomotor activity *in vitro*. In our previous investigations, we also found that interlimb coordination in adult rats was affected in a short time (<5 min) after the intrathecal application of SB269970 [32]. These early drug effects on left-right coordination might be related to interfering with short ascending commissural connections. While the effect described in the present paper in the time of ELRP (>20 min), the left-right coordination is not any more affected, there is a clear effect on fore-hind limb coupling due to 5-HT_7_ receptor modulations.

There is very limited data available about 5-HT_7_ receptor localization in the rat spinal cord. Doly with coworkers focused their attention on the dorsal spinal cord and localized 5-HT_7_ receptors mainly in the laminae I-III of the lumbar level in rats [69]. They also observed some low labeling in the ventral horn on motoneurons in lamina IX suggesting that 5-HT_7_ might be involved in motoneuron control, as was suggested by Schmidt and Jordan [28,69]. There is more data from cats supporting our hypothesis. Noga with coworkers [70] identified serotonergic 5-HT_7_ receptors on the soma and proximal dendrites of neurons activated (defined by upregulation of c-fos expression) during fictive locomotion within laminae VII and VIII of thoracolumbar segments. The immunoreactive cells were surrounded by serotonergic fibers, which appeared to make contact with these cells. The hypothesis about the functional role of lumbar ascending propriospinal connection in control forelimb locomotor pattern is supported by our results indicating 5-HT_7_ immunoreactivity present in the lamina VII/VIII, where cell bodies of long ascending propriospinal interneurons are located [40], and which were affected by SB269970 but not by cyproheptadine. The 5-HT_2A_ immunoreactivity, very rarely observed in the lamina VII, and extensively present on the big cells in lamina IX (motoneuron area) in the lumbar ventral horn is explaining the differences in the effects of both drug applications. All these data support our hypothesis that the 2FL-1HL coupling effect during recovery of quadrupedal locomotion after SB269970 application might be related to the 5-HT_7_ modulation of long ascending propriospinal interneurons located in laminae VII/VIII in the lumbar spinal cord segments. It might be also considered that a particular balance of 5-HT_7_ receptor modulation might be required for such locomotor effect. Our investigation demonstrated that cyproheptadine (5-HT_2A_ antagonist) did not induce such locomotor effect, even though it is known that it also presents some affinity to 5-HT_7_ receptors [71]. However, its affinity to 5-HT_7_ receptors is more than 20 times smaller than that of SB269970 (https://pdsp.unc.edu/databases/pdsp.php, accessed on 26 April 2021). Thus, in addition to the different laminae distribution of cells expressing 5-HT_2A_ and 5-HT_7_ for the absence of the 2FL-1HL pattern after cyproheptadine, the predominance of the 5-HT_2A_ blockade effect but not an effect of 5-HT_7_ receptor blockade in the lumbar spinal cord might be responsible (15 times higher affinity of cyproheptadine to 5-HT_2A_ than to 5-HT_7_ receptors).

### 3.5. 5-HT Sensory Feedback Control

It is known that the modulation of spontaneous unrestrained locomotion not only requires feedforward supraspinal but also feedback sensory control [72,73,74,75,76,77,78,79,80]. Our previous investigations demonstrated that the sensory feedback from leg proprioceptive receptors controlling locomotor performance is under serotonergic control [32,81]. 5-HT control of muscle [82,83,84] and cutaneous afferents [85] is well documented. 5-HT_7_ receptors have been identified on dorsal root ganglion cells as well as on cells in dorsal horns (laminae I and II) of the lumbar segments [69,85]. 5-HT_7_ receptors localized in the dorsal horn are involved in the processing of sensory information [70,85,86]. Presynaptic or postsynaptic inhibitory control of afferent input during locomotion by 5-HT_7_ receptor-mediated control of GABAergic interneurons in the spinal cord was also suggested [31,69]. Consistent with an antinociceptive action mediated by 5-HT_7_ receptors is also discomfort that we observed in intact rats after intrathecal SB269970 in our initial experiments (see Methods). Thus, the 2FL-1HL locomotor pattern obtained in our investigations might be also an effect of the 5-HT_7_ mechanism that integrates various modality inputs from descending as well as proprioceptive and nociceptive afferents in the lumbar spinal cord networks modulating locomotor performances in freely moving rats. Further work is needed to investigate whether spinal sensorimotor interaction at the lumbar ascending projection might be responsible for such effects described in the present paper.

## 4. Materials and Methods

Wistar female rats 3-months-old with a bodyweight of 242 ± 23 g at the beginning of the experiments were used in our investigations (*n* = 8). Experimental procedures and all surgical actions were carried out with care to minimize pain and suffering of animals, and were approved by the First Ethics Committee for Animal Experimentation in Poland (decision no. 85/2010 and 636/2014), according to the principles of experimental conditions and laboratory animal care of the European Union (EU Directive, 2010/63/EU) and the Polish Law on Animal Protection.

### 4.1. Intrathecal Cannula Implantation

The implantation of the intrathecal cannula was performed in aseptic conditions under deep anesthesia (isoflurane 2%; butomidor 0.05 mg/kg b.w.) as previously described [32,33]. First, the skin was transected over Th10/11 vertebral spines and after the back muscle separation, the cannula (polyethylene PE-10) was inserted into the subarachnoid space between these two spinal vertebrae through a small opening and pushed under the dura caudally aiming to reach the lumbar segments of the spinal cord (Th13-L2 vertebrae). Next, the cannula was fixed to the Th10 spinous process by a suture (5/0). Closing by suturing the overlying back muscles supports additionally the cannula stabilization. A custom-made adaptor with the other end of the cannula connected was fixed by dental cement (Spofa Dental, Prague, Czechia) to the skull. After the cannula implantation, the rats received a non-steroidal anti-inflammatory and analgesic treatment (tolfedine 4 mg/kg s.c.) and antibiotic (baytril 5 mg/kg s.c.). In the following days after surgery (two to five) the cannula patency was verified by injection of 15 μL of 2% lidocaine followed by 12 μL of sterile saline (NaCl 0.9%).

### 4.2. Implantation of EMG Electrodes

To record the EMG activity in freely moving rats the bipolar electrodes were implanted into the selected muscles of each limb (in the forelimbs triceps brachii (Tri) muscle and the hindlimbs soleus (Sol) and tibialis anterior (TA) muscles) under deep anesthesia (isoflurane, 2% and butomidor, 0.05 mg/kg b.w.) as we described previously [32,33]. The homemade electrodes for chronic EMG recording were manufactured using the Teflon-coated multi-stranded stainless steel wire (0.24 mm in diameter; AS633, Cooner Wire, Chatsworth, CA, USA). The stainless steel wires with the hook as the electrodes (the 1–1.5 mm of the insulation removed) were inserted into the appropriate muscle with a distance of 1–2 mm and secured by a suture after pulling under the skin through a cutaneous incision on the back of the animal (as it was described in [32,33]). The other ends of the wires were secured to the connector that was covered with dental cement (Spofa Dental) and silicone (3140 RTV, Dow Corning, Midland, MI, USA). The connector was placed and secured by sutures under the skin of the animal back simultaneously with the ground electrode at some distance from the hindlimb muscles. After completing electrode implantation the rats were given a non-steroidal anti-inflammatory and analgesic treatment (tolfedine, 4 mg/kg s.c.) and antibiotic (baytril, 5 mg/kg s.c.).

### 4.3. Testing the Effects of Intrathecal Drug Applications

To block 5-HT_2A_ receptors we used their inverse agonist cyproheptadine chloride (Table 4; 4-(5*H*-dibenzo[a,d] cyclohepten-5-ylidine)-methylpiperidine hydrochloride) with a dose of 150 μg in 20 μL solution. To block 5-HT_7_ receptors we used their antagonist -SB269970 (Table 4; (2*R*)-1-[(3- hydroxyphenyl)sulfonyl]-2-[2-(4-methyl-1-piperidinyl) ethyl] pyrrolidine hydrochloride) with dose 150 μg in 30 μL of saline. Cyproheptadine was dissolved for a stock solution in glycerol (plus 10 μL 0.1N HCl). The final dose was prepared from the stock solution by adding 0.9% NaCl just before intrathecal application. To reduce animal discomfort during intrathecal drug application, each experiment started with a subcutaneous application of butomidor (0.1 mg/kg, S.C.; butorphanol; Richter Pharma, Wels, Austria), which is a synthetic opioid agonist/antagonist with a potent analgesic and moderate sedative properties. Butomidor subcutaneous application by itself did not alter the treadmill locomotor pattern that was shown in control experiments. In the separate control experiment, the vehicles used for the preparation of drug suspension were injected alone, and none of them induced any changes in the locomotor performance of tested animals.

### 4.4. Locomotor Performance and EMG Recordings

To monitor four-limb locomotor movements the EMG activity of defined muscles was recorded. The locomotion was tested on a treadmill (Panlab/Harvard Apparatus, Barcelona, Spain). Rats were trained to perform locomotor movements at stable rhythmicity at the defined speed of a belt. The quadrupedal locomotor performance was investigated on a treadmill with simultaneous recording of EMG activity of selected fore and hind leg muscles before and after intrathecal drug application during a locomotor performance at the rats’ preferred speed ranging between 10 to 20 cm/s. Care was taken to record locomotor behavior before and after drug application at the same treadmill speed. In all the experiments only runs with at least 15 regular steps were accepted for further analysis. After evaluation of pre-drug locomotor performance, a bolus of 20 or 30 μL of selected antagonist suspension was delivered through the intrathecal cannula (in 60 s) that was followed immediately by a bolus of 12 μL of saline to wash out the drug from the cannula. The drug effect described in this manuscript was tested at the time when animals regained four limb locomotion with plantar stepping in the time of ELRP (median time after SB269970 was 21 min; after cyproheptadine was 40 min). The short time effect with total hindlimb paralysis induced by intrathecal application of the same drugs (SB269970 or cyproheptadine) at the level lumbar spinal cord of the same Wistar adult rats was described in our previous papers [32,33] and is not within the scope of the present manuscript.

### 4.5. Locomotor EMG Pattern Analysis

The locomotor pattern analysis was carried out using rhythmic EMG burst activity of the flexor and extensor muscles of the limbs of the front and hind girdles recorded during spontaneous uninterrupted locomotion on a treadmill. The recorded EMG signals were filtered (0.1- to 1 kHz bandpass), digitized, and stored on a computer (2 kHz sampling frequency) for offline analysis.

First, forelimb–hindlimb coupling was assessed by establishing the Regularity Index (RI) that we defined as the ratio of the number of steps performed by forelimb (Tri EMG burst activity) to the number of steps performed by hindlimb (Sol EMG burst activity) during spontaneous uninterrupted locomotion on a treadmill. The RI value close to 1 describes step sequence patterns during sustained, coordinated locomotion, whereas impairment of fore-hindlimb coordination leads to missteps interspersed between regular steps during the quadrupedal locomotor performance, causing the RI value to increase in the case of more steps taken by forelimbs or decrease in the case fewer steps taken by forelimbs in relation to the number of steps taken by hindlimbs [42,45].

Next, the EMG pattern was analyzed using the Winnipeg Spinal Cord Research Centre data capture and analysis software (http://www.scrc.umanitoba.ca/doc/, accessed on 26 April 2021). At the beginning of the analysis, the raw EMG signals were rectified and integrated with 5 ms intervals. Next, marking all the burst onsets and offsets in the EMG activity recorded in various muscles within identified fragments of the locomotor EMG recordings with 15–20 steps in different experimental conditions was performed. The marked onsets and offsets of consecutive EMG bursts allowed us to establish the cycle duration, burst EMG amplitude, and interlimb coordination for locomotor patterns of different rats before and after drug application. The marked onsets and offsets of consecutive EMG bursts allowed us to establish the EMG envelopes of each analyzed muscle activity. To do this, the rectified and filtered EMG records were normalized to each single step cycle, taking the onset of burst EMG activity in the Tri muscle as the onset of the cycle and showing the ipsilateral envelopes for Sol and TA over the same step cycle displayed. To investigate the phase shift distribution of the consecutive steps taken by two different limbs before and after drug administration, we used a circular analysis method. The left-right interlimb coordination (the coupling of homonymous muscles of the left and right limb) of both girdles as well as interlimb coordination between ipsilateral limbs (muscles of the fore vs. hind limbs on the same body side) or diagonal limbs (muscles of the fore and hind limb on the opposite body side), were determined using polar plot analysis [12,33,87,88,89]. The polar plot analysis gives results in a form of an *r*-vector with a particular length (*r*-value, ranging from 0 to 1) and angle position at the circle (in the range from 0° to 360°). The vector position at 0° or 360° of the circular plot reflects perfect synchrony of analyzed EMG burst onsets, whereas 180° is equivalent to alternation. The *r*-value (vector length) reflects the concentration of phase shift values around the mean indicating the strength of coordination between analyzed muscle burst onsets (the dots at the circle). The phase shift values (individual dots on the polar plot) should be highly concentrated around the mean phase when the onset of EMG bursts of the analyzed muscles are strongly coupled. When there is no coupling in analyzed onsets of EMG bursts of analyzed muscle activities, the dots on the polar plot show some dispersion with wide distribution. Rayleigh’s circular statistical test was used to determine whether *r*-values related to the interlimb coupling were significant. The interlimb coordination was considered significant (phase-related) when the *r*-value was greater than the critical Rayleigh’s value (cR) for a given *p*-value [88].

### 4.6. Statistical Analysis

All data are reported as mean ± SD (standard deviation). For all statistical tests used in this study *p*-value ≤ 0.05 was considered to be significant. For phase comparisons of the circular data, Watson-Williams angular test was used, a circular analog of the one-factor ANOVA (IGOR Pro software; Wavemetrics, Lake Oswego, OR, USA). The other sets of data were analyzed using GraphPad software for Windows (San Diego, CA, USA). One-sample Wilcoxon test (a non-parametric statistical hypothesis test) was used for checking if post-drug data expressed as a ratio to pre-drug values were different from 1. For statistical testing of the application effects of both drugs, when the post-drug data were expressed as a ratio to pre-drug ones, non-parametric Kruskal-Wallis (KW) statistical analysis was applied. The same statistical procedure was used for comparison of the strength of interlimb coordination established as the length of the *r*-vector in different experimental conditions. The results of KW non-parametric statistical analysis are provided in the text describing results illustrated in the particular figure as KW statistics value (with degrees of freedom in parentheses) and the corresponding *p*-value. In the case of multiple comparisons, the significance of the *p*-value was additionally checked with Benjamini-Hochberg 5% false discovery rate (FDR) procedure [90] According to a suggestion given by McDonald [91] we did not present FDR corrected *p*-values. Instead, we show the original *p*-values and describe these which remain significant after using the Benjamini-Hochberg FDR procedure. Sign test was calculated online using the “Social Science Statistics” web page (https://www.socscistatistics.com, accessed on 26 April 2021).

### 4.7. Immunohistochemistry

After completing the functional investigations the rats were perfused with cold 0.1 M PBS and 4% paraformaldehyde (PFA) in PBS under deep anesthesia (induced by intraperitoneal injection of sodium pentobarbital, 150 mg/kg). The dissected spinal cords were postfixed overnight in 4% PFA and cryoprotected gradually by 10, 20, and 30% sucrose in PBS. For cryostat cutting, spinal cords were first embedded in an optimal cutting temperature compound (OCT embedding matrix, CellPath, Newton, Powys, UK) and then frozen on dry ice. The spinal cord was cut into slices (12 µm), immobilized on the poly-l-Lys-covered glass and stored at −80 °C before use.

Sections were first washed in PBS several times and placed in 1% H_2_O_2_ for 10 min to quench endogenous peroxidase activity. Next, the sections were blocked in 10% serum in PBS with 0.5% Triton X-100 at room temperature (RT) for 1 h. After 96 h incubation at 4 °C in PBS with 0.1% Triton X-100, 2% serum, and primary antibody (mouse anti-5-HT_2A_ antibody, 1:25, Merck, Darmstadt, Germany or rabbit anti-5-HT_7_ primary antibody, 1:75, ImmunoStar, Hudson, WI, USA). Next, sections were incubated with biotinylated anti-mouse antibody (1:200 in PBS, Vector Laboratories, Burlingame, CA, USA or biotinylated anti-rabbit secondary antibody (1:200, Vector Laboratories) for 1 h at RT. Afterward, sections were rinsed in PBS. Next, they were treated with Vectastain Elite ABC Reagent and then exposed to DAB Peroxidase Substrate (Vectastain Elite ABC Kit, DAB Substrate Kit; Vector Laboratories), according to the manufacturer protocol. Finally, after dehydration, the slides were coverslipped with DPX new (Merck). For 5-HT_7_ staining, all steps were conducted without Triton X-100. For final examinations and image acquisition, Axio Imager M2 (Zeiss, Oberkochen, Germany) and Eclipse Ts2R (Nikon, Tokyo, Japan) microscopes were used.

## 5. Conclusions

Our results show for the first time that blockade of 5-HT_7_ receptors in the lumbar spinal cord of adult rats facilitates in the time of the ELRP a double rhythm in the forelimbs in relation to the hindlimb locomotor pattern (2FL-1HL fore-hindlimb) during quadrupedal locomotion on the treadmill, in contrast to the absence of such effect after blockade of 5-HT_2A_ receptors. We showed that during the episodes of analyzed quadrupedal walking with the 2FL-1HL coupling pattern the forelimb cycles remain locked with the hindlimb cycles. This indicates that in the time of the ELRP a new form of fore-hindlimb coordination can be established as an effect of 5-HT_7_ receptors’ modulation by intrathecal application of SB269970. These results indicate that 5-HT_7_ receptors are crucial for coordination of the fore- and hindlimb CPG networks that control quadrupedal locomotion. We suggest that for the emergence of such a new stable state of the four-limb CPG network coupled activity some ascending propriospinal interneurons of the lumbar cord projecting to the cervical network to control the forelimb movements with 5-HT_7_ but not 5-HT_2A_ receptors might be responsible. This prediction should be tested using 5-HT_7_ receptor knockout models.

## Figures and Tables

**Figure 1 ijms-22-06007-f001:**
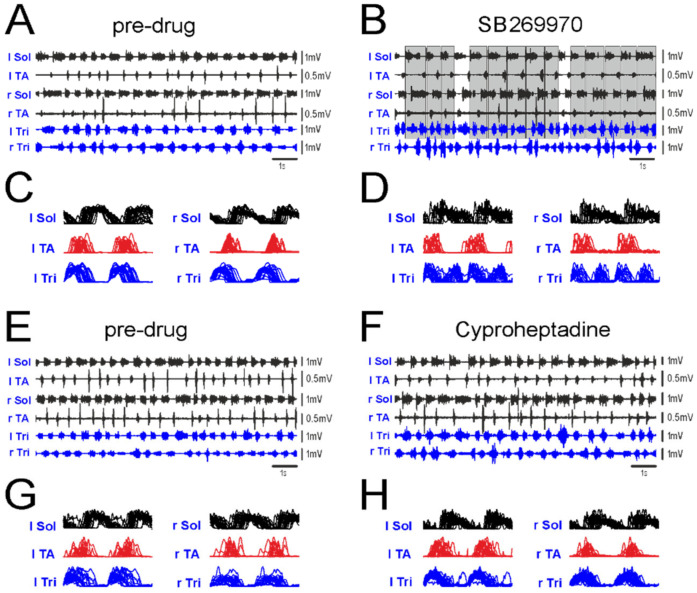
Examples of EMG activity recorded from hindlimb (Sol, TA) and forelimb (Tri) muscles in the same rat during locomotion on a treadmill (15 cm/s) before and after blockade of 5-HT_7_ or 5-HT_2A_ receptors by intrathecal application of SB269970 (**A**–**D**) or cyproheptadine (**E**–**H**) at lumbar spinal cord level. There is an obvious 2FL-1HL stepping pattern after the blockage of 5-HT_7_ receptors by SB269970 (**B**,**D**), but not after blockade of 5-HT_2A_ receptors by cyproheptadine (**F**,**H**). Note, that the 2FL-1HL locomotor pattern can be interspersed with normal 1FL-1HL patterns (**B**). Gray rectangles indicate all the cycles in the left Sol EMG with double bursting in the left Tri EMG muscle. The linear envelopes (**C**,**D**,**G**,**H**) of rectified and integrated EMG activity (superimposed envelopes repeated twice to visualize two cycles) established for the EMG activities illustrated in (**A**,**B**,**E**,**F)** demonstrate clear two shorter forelimb cycles in the Tri burst EMG activity of the 2FL-1HL pattern related to one cycle in the EMG of Sol and TA on the left as well as on the right pair of legs after SB269970 application (**D**) while in pre-drug (**C**,**G**) and after cyproheptadine (**H**) clear 1FL-1HL pattern is present. *Abbreviations*: l Sol/r Sol, left/right soleus muscle; l TA/r TA, left/right tibialis anterior muscle; l Tri/r Tri, left/right triceps brachii muscle.

**Figure 2 ijms-22-06007-f002:**
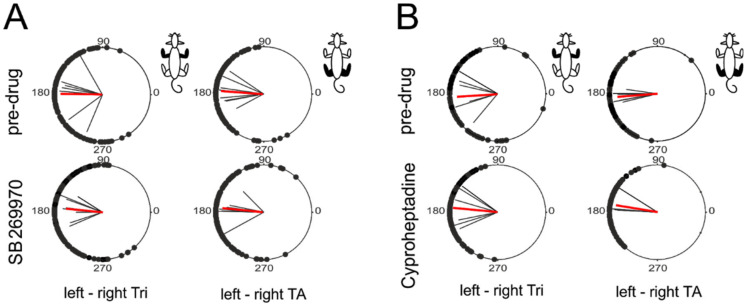
Polar plots showing clear interlimb left-right coordination in fore- as well as hindlimbs during treadmill quadrupedal locomotion (with the belt speed 10–20 cm/s) of adult rats before (pre-drug) and during recovery from blockade of 5-HT_7_ or 5-HT_2A_ receptors. Blocking the corresponding receptors was caused by the intrathecal application of SB269970 (**A**) or cyproheptadine (**B**). Post-drug data were collected during the ELRP (early locomotor recovery period). Black dots on each circle represent the phase shift between corresponding onsets of EMG burst of left vs. right Tri and left vs. right TA in the analyzed individual steps in the appropriate group of animals, black straight lines indicate mean coordination *r*-vectors for individual animals and red straight lines mean coordination *r*-vectors for the groups of animals (in each case, the length of the *r*-vector represents the mean coordination strength and the *r*-vector angle indicates the mean coordination phase shift). Watson Williams test for phase comparisons did not show any statistically significant differences between pre- and post-drug conditions. The limb pairs analyzed in each panel are shown in black in the rat diagram in the upper right corner. *Abbreviations*: TA, tibialis anterior muscle; Tri, triceps brachii muscle.

**Figure 3 ijms-22-06007-f003:**
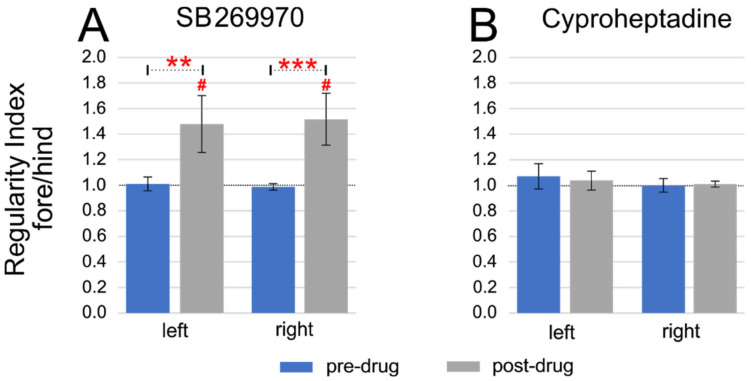
Regularity Index (RI) established as the number of steps performed by forelimb (triceps EMG burst activity) in ratio to the number of steps performed by hindlimb (soleus EMG burst activity) during spontaneous locomotion on a treadmill in intact rats after blockage of 5-HT_7_ (**A**) or 5-HT_2A_ (**B**) receptors by intrathecal application of SB269970 or cyproheptadine (respectively) at lumbar spinal cord segments. Statistical significance: ** *p* < 0.005; *** *p* < 0.001; Statistical significance for difference from 1: ^#^
*p* < 0.05.

**Figure 4 ijms-22-06007-f004:**
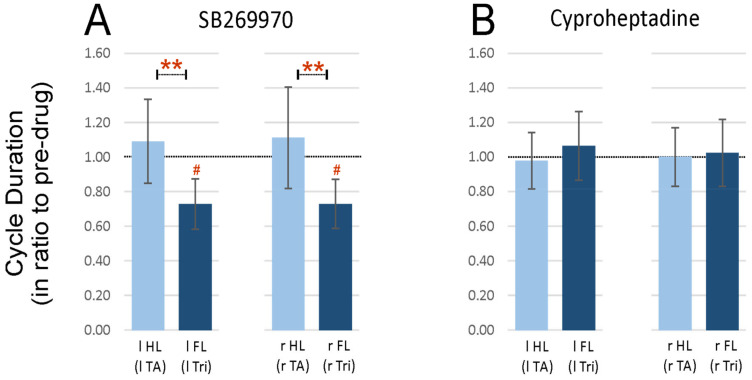
Cycle duration established in forelimbs (based on the Tri EMG activity) and hindlimbs (based on the TA EMG activity) in ratio to that from the pre-drug condition during spontaneous locomotion on a treadmill in intact rats after blockade of 5-HT_7_ (**A**) or 5-HT_2A_ (**B**) receptors by intrathecal application of SB269970 or cyproheptadine (respectively) at lumbar spinal cord segments. Statistical significance for comparison between fore vs. hind limb cycles: ** *p* < 0.005; Statistical significance for difference from 1. ^#^
*p* < 0.05. *Abbreviations*: TA, tibialis anterior muscle; Tri, triceps brachii muscle; l, left; r, right; FL, forelimb; HL, hindlimb.

**Figure 5 ijms-22-06007-f005:**
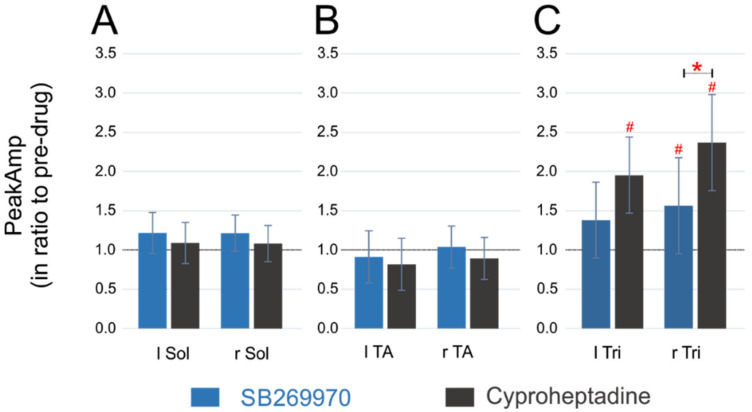
The peak amplitude of the EMG burst activity in ratio to the pre-drug condition established in the Tri muscle (**C**) of both forelimbs and the Sol (**A**) and TA (**B**) muscles of both hindlimbs during spontaneous locomotion on a treadmill in adult rats after blockade of 5-HT_7_ or 5-HT_2A_ receptors by intrathecal application of SB269970 or cyproheptadine (respectively) at lumbar spinal cord segments. Statistical significance for comparison of effects of both drugs: * *p* < 0.05; Statistical significance for difference from 1: ^#^
*p* < 0.05.

**Figure 6 ijms-22-06007-f006:**
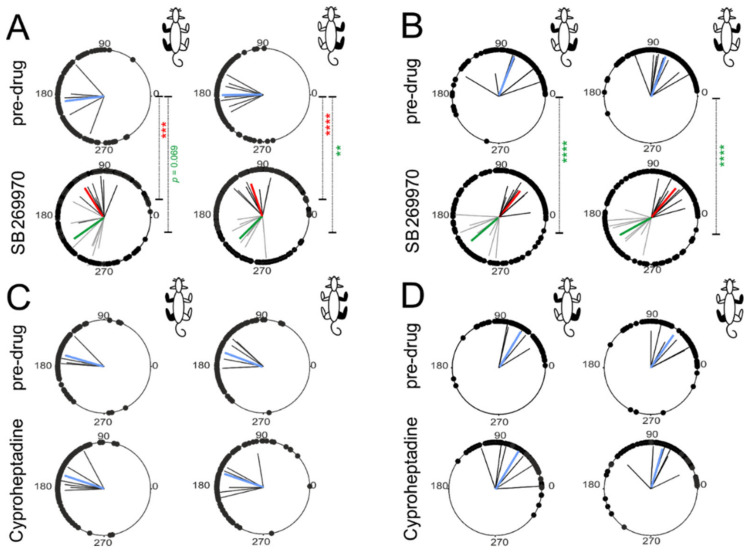
Polar plots showing ipsilateral (between forelimb Tri and ipsilateral hindlimb Sol muscles) and diagonal (between forelimb Tri and diagonal hindlimb TA muscles) interlimb coordination during treadmill quadrupedal locomotion (with the belt speed 10–20 cm/s) of intact rats before (pre-drug) and during recovery from blockade of 5-HT_7_ or 5-HT_2A_ receptors. Blocking the corresponding receptors was caused by the intrathecal application of SB269970 (**A**,**B**) or cyproheptadine (**C**,**D**). Post-drug data were collected during the ELRP (early locomotor recovery period). Black dots on each circle represent the phase shift between corresponding EMG bursts of defined above muscles for ipsilateral and diagonal coordination in the analyzed individual steps in the appropriate group of animals. In the cases of the 1FL-1HL pattern (both pre-drug conditions and ELRP after cyproheptadine application), black straight lines indicate the mean coordination *r*-vectors for individual animals, while blue straight lines mean coordination *r*-vectors for the groups of animals. In the case of the 2FL-1HL pattern (ELRP after SB269970 application), black straight lines indicate the mean “1st Tri burst” coordination *r*-vectors for individual animals (and then red straight lines represent mean coordination *r*-vectors for the group of animals), while gray straight lines indicate the mean “2nd Tri burst” coordination *r*-vectors in individual animals (and then green straight lines represent the mean coordination *r*-vectors for the group of animals). In each case, the length of the *r*-vector represents the mean coordination strength and the *r*-vector angle indicates the mean coordination phase shift. Watson Williams test for phase shift comparisons: ** *p* < 0.005; *** *p* < 0.001; **** *p* < 0.0001 (the red or green color of asterisks indicating the level of statistical significance is adequate to the color of the mean *r*-vector representing the cluster of post-drug data with which the pre-drug condition is compared). The limb pairs analyzed in each panel are shown in black in the rat diagram in the upper right corner. *Abbreviations*: TA, tibialis anterior muscle; Tri, triceps brachii muscle.

**Figure 7 ijms-22-06007-f007:**
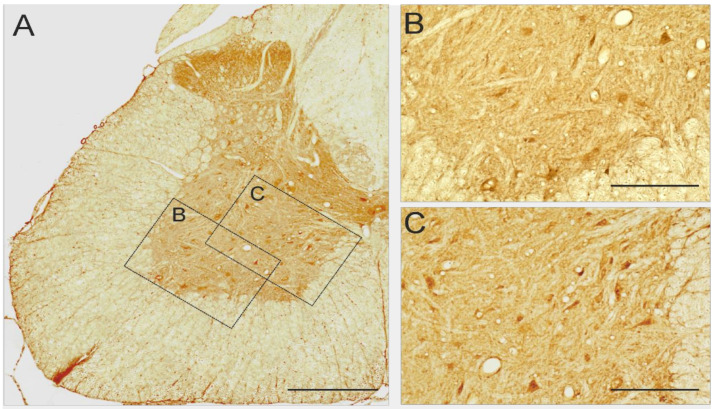
5-HT_7_ receptor immunoreactivity in the lumbar spinal cord in adult rat (**A**), on small cells in the lamina IX (**B**), and on small cells in the lamina VII and VIII (**C**). 5-HT_2A_ receptors in the lumbar spinal cord in adult rat (**D**) on big and small cells in the lamina IX (**E**) and hardly any staining in the lamina VII (**F**). Scale bars: 500 μm (**A**,**D**); 200 μm (**B**,**C**,**E**,**F**).

**Table 1 ijms-22-06007-t001:** Interlimb coordination in pairs of limbs of fore and hind girdles during quadrupedal locomotion before and after SB269970 or cyproheptadine application.

		SB269970		Cyproheptadine	
		Forelimbs	Hindlimbs	Forelimbs	Hindlimbs
		Left-Right Tri	Left-Right TA	Left-Right Tri	Left-Right TA
*r*-Value	pre-drug	0.85 ± 0.11	0.85 ± 0.10	0.82 ± 0.12	0.82 ± 0.11
	post-drug	0.75 ± 0.13	0.82 ± 0.12	0.91 ± 0.05	0.85 ± 0.15
	*p*-value	0.10	0.61	0.19	0.29
Phase (°)	pre-drug	179.0 ± 38.3	177.8 ± 20.2	183.6 ± 30.0	184.5 ± 8.3
	post-drug	174.3 ± 19.6	174.2 ± 20.3	174.5 ± 26.3	170.6 ± 11.6
	*p*-value	0.81	0.73	0.61	0.039

**Table 2 ijms-22-06007-t002:** Ipsilateral and diagonal limb coordination during quadrupedal locomotion before and after SB269970 application.

		SB 269970			
		Ipsilateral		Diagonal	
		Left Tri-Sol	Right Tri-Sol	Left Tri-Right TA	Right Tri-Left TA
R-Value 1st Tri Burst	pre-drug	0.83 ± 0.11	0.84 ± 0.06	0.87 ± 0.16	0.89 ± 0.05
	post-drug	0.71 ± 0.14	0.72 ± 0.11	0.70 ± 0.12	0.78 ± 0.15
	*p*-value	0.063	0.034	**0.006**	0.16
Phase (°) 1st Tri Burst	pre-drug	187.3 ± 33.0	181.0 ± 19.4	68.9 ± 40.9	70.5 ± 26.0
	post-drug	115.7 ± 37.5	109.3 ± 22.7	51.4 ± 21.5	49.9 ± 20.8
	*p*-value	**0.00099**	**0.00005**	0.37	0.12
r-Value 2nd Tri Burst	pre-drug	0.83 ± 0.11	0.84 ± 0.06	0.87 ± 0.16	0.89 ± 0.05
	post-drug	0.76 ± 0.10	0.68 ± 0.14	0.75 ± 0.19	0.76 ± 0.18
	*p*-value	0.13	**0.008**	0.085	0.088
Phase (°) 2nd Tri Burst	pre-drug	187.3 ± 33.0	181.0 ± 19.4	68.9 ± 40.9	70.5 ± 26.0
	post-drug	220.7 ± 38.8	224.9 ± 27.0	220.6 ± 33.0	208.9 ± 31.2
	*p*-value	0.069	**0.003**	**0.00005**	**0.00005**

*p*-values that remained significant after 5% FDR correction for multiple comparisons are written in bold.

**Table 3 ijms-22-06007-t003:** Ipsilateral and diagonal limb coordination during quadrupedal locomotion before after cyproheptadine application.

		Cyproheptadine			
		Ipsilateral		Diagonal	
		Left Tri-Sol	Right Tri-Sol	Left Tri-Right TA	Right Tri-Left TA
*r*-Value	pre-drug	0.83 ± 0.16	0.84 ± 0.04	0.85 ± 0.13	0.83 ± 0.12
	post-drug	0.87 ± 0.08	0.87 ± 0.1	0.94 ± 0.01	0.84 ± 0.12
	*p*-value	0.93	0.29	0.18	0.84
Phase (°)	pre-drug	163.9 ± 15.2	159.5 ± 21.5	59.2 ± 17.5	54.9 ± 25.5
	post-drug	161.7 ± 24.2	161.2 ± 34.6	58.7 ± 38.0	74.0 ± 35.7
	*p*-value	0.88	0.83	0.98	0.33

**Table 4 ijms-22-06007-t004:** Chemical agents and their affinity (in nM) to different 5-HT receptors used for the intrathecal application.

Agent	5-HT_2A_	5-HT_2B_	5-HT_2C_	5-HT_7_
SB269970	Ki > 10,000	Ki > 10,000	Ki > 10,000	Ki = 0.34–1258
Cyproheptadine	Ki = 0.44	Ki = 1.54	Ki = 2.23	Ki = 7.5 ^#^

https://pdsp.unc.edu/databases/pdsp.php (accessed on 26 April 2021). ^#^ [71].

## Data Availability

All data used in this study are included in the article. Any additional information required will be provided upon request to the corresponding author.

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
