# Peer review of "Unusual Quadrupedal Locomotion in Rat during Recovery from Lumbar Spinal Blockade of 5-HT7 Receptors"

_ijms, 2021, doi:10.3390/ijms22116007_

Round 1

Reviewer 1 Report

  1. Recommendation:

Minor Revision

  1. Comments to Author:

Manuscript ID: ijms-1217663

Title: Unusual quadrupedal locomotion in the rat during recovery from lumbar spinal blockade of 5-HT7 receptors

 I found the paper to be perfectly written and all the sections described in detail. All results obtained in the study are elegantly  illustrated in figures and tables, and full information of results is provided in the discussion section. For all these reasons, I consider that the current study is relevant and of interest to the readers of the journal.

  • Minor comments:

Line 249: ‘the results presented here’ instead of ‘the presented here results’

Figure 5 legend: Meaning for A, B and C is missing. I would recommend to add it in the legend as follows: The peak amplitude of the EMG burst activity in ratio to the pre-drug condition established in the Tri muscle (C) of both forelimbs and the Sol (A) and TA (B) muscles of both hindlimbs…

Author Response

  1. Recommendation:

Minor Revision

  1. Comments to Author:

Manuscript ID: ijms-1217663

 Title: Unusual quadrupedal locomotion in the rat during recovery from lumbar spinal blockade of 5-HT7 receptors

  I found the paper to be perfectly written and all the sections described in detail. All results obtained in the study are elegantly illustrated in figures and tables, and full information of results is provided in the discussion section. For all these reasons, I consider that the current study is relevant and of interest to the readers of the journal. 

We are very pleased with the effort taken by the Reviewer. We have corrected our manuscript according to the Reviewer’s suggestions.

Minor comments:

Line 249: ‘the results presented here’ instead of ‘the presented here results’

Corrected.

Figure 5 legend: Meaning for A, B and C is missing. I would recommend to add it in the legend as follows: The peak amplitude of the EMG burst activity in ratio to the pre-drug condition established in the Tri muscle (C) of both forelimbs and the Sol (A) and TA (B) muscles of both hindlimbs…

Corrected.

Reviewer 2 Report

Coordination of forelimbs and hindlimbs during locomotion involves circuits that span cervical and hindlimb spinal segments.  Several studies suggest asymmetry in the connections from forelimb to hindlimb and those connecting hindlimbs with forelimbs.  

This study analyzes locomotor function in behaving rats after temporary paralysis induced by application of a 5-HT2 and 5-HT7 antagonist to show that recovery after a blockage of 5-HT7 receptors leads to an unusual doubling of forelimb stepping in relation to hindlimb stepping. 

EMG recordings of forelimb and hindlimb muscles were made during locomotor function after the temporary paralysis. Analysis of limb coordination and rhythmic activity during recovery from 5-HT2 versus 5-HT7 antagonism is methodical and rigorous. Evidence points to disturbance of forelimb to hindlimb coordination during recovery from 5-HT7 but not 5-HT2 antagonism. 

My only issue is that the immunohistochemistry staining is not the most convincing.  Additional staining of motoneurons and retrograde staining of ascending lumbar propriospinal neurons would really strengthen the findings of this study.

Line 117, were the 6 out of 8 animals that showed effects, the same 6 out of 8 rats that were tested for both cyproheptadine and SB269970 referred to at line 107? 

Line 123-125. A bit confusing. Do you mean to say something like: "The 2FL-1HL patterns were sometimes interspersed with the normal 1FL-1H. The analysis of locomotor activity was performed on epochs where the 2FL-1HL patterns were stable"?

Lines 421-426. This sentence could be clarified.  I suggest: "Although significantly lower r-vector values were obtained during the post-drug condition in 2 individual coordinations, all 10 r-vector values for interlimb coordination in each girdle as well as ipsilateral and diagonal coordiations between girdles established after application of 5-HT7 receptor antagonist were decreased compared to pre-drug conditions.

Is there any chance you can do concurrent ChAT staining for motoneurons. The images shown are not convincing in terms of selective staining onto motoneurons.  

The immunohistochemistry experiments would provide stronger support if it could be shown that 5-HT7 expressing neurons in the lumbar region were long ascending propriospinal neurons. Would it be possible to perform retrograde staining of lumbar ascending propriospinal neurons and show that these cells express 5-HT7 receptors but not 5-HT2 neurons?

Author Response

Coordination of forelimbs and hindlimbs during locomotion involves circuits that span cervical and hindlimb spinal segments.  Several studies suggest asymmetry in the connections from forelimb to hindlimb and those connecting hindlimbs with forelimbs.  

This study analyzes locomotor function in behaving rats after temporary paralysis induced by application of a 5-HT2 and 5-HT7 antagonist to show that recovery after a blockage of 5-HT7 receptors leads to an unusual doubling of forelimb stepping in relation to hindlimb stepping. 

EMG recordings of forelimb and hindlimb muscles were made during locomotor function after the temporary paralysis. Analysis of limb coordination and rhythmic activity during recovery from 5-HT2 versus 5-HT7 antagonism is methodical and rigorous. Evidence points to disturbance of forelimb to hindlimb coordination during recovery from 5-HT7 but not 5-HT2 antagonism. 

We are very pleased with the effort taken by the Reviewer. We corrected our manuscript according to the Reviewer’s suggestions. We hope our corrections improved the text of our manuscript and made it clear now.

My only issue is that the immunohistochemistry staining is not the most convincing. Additional staining of motoneurons and retrograde staining of ascending lumbar propriospinal neurons would really strengthen the findings of this study.

We agree with the Reviewer that motoneuron identity was not supported by our results. Due to the specific nature of receptor immunostaining that required preservation of cell membrane integrity and technical specification of the available antibodies we were not able to perform double immunostaining for a particular receptor together with motoneuron markers. Thus we took care to modify the results description and figure legend regarding this issue by pointing that in the ventral horn there we identified immunopositive big cells that presented motoneuron phenotype.

Line 117, were the 6 out of 8 animals that showed effects, the same 6 out of 8 rats that were tested for both cyproheptadine and SB269970 referred to at line 107? 

This sentence was removed and the issue was clarified in the foregoing sentence, which sounds now as follows: “In contrast, when 5-HT2A receptors were blocked by Cyproheptadine application in the 6  rats (out of 8 tested with SB269970) the four-limb locomotor pattern remained unaffected: the locomotor pattern in forelimbs remained coupled to that of hindlimbs (1FL-1HL). “

Line 123-125. A bit confusing. Do you mean to say something like: "The 2FL-1HL patterns were sometimes interspersed with the normal 1FL-1H. The analysis of locomotor activity was performed on epochs where the 2FL-1HL patterns were stable"?

This sentence was clarified and sounds now as follows: “Although in the analyzed episodes of locomotor activity the fragments of the 2FL-1HL patterns were stable and consistent, occasionally, they could be interspersed with single-stepping of the normal 1FL-1HL pattern (see example in Fig. 1 B).

Lines 421-426. This sentence could be clarified.  I suggest: "Although significantly lower r-vector values were obtained during the post-drug condition in 2 individual coordinations, all 10 r-vector values for interlimb coordination in each girdle as well as ipsilateral and diagonal coordiations between girdles established after application of 5-HT7 receptor antagonist were decreased compared to pre-drug conditions.

This sentence was corrected.

Is there any chance you can do concurrent ChAT staining for motoneurons. The images shown are not convincing in terms of selective staining onto motoneurons.  

The immunohistochemistry experiments would provide stronger support if it could be shown that 5-HT7 expressing neurons in the lumbar region were long ascending propriospinal neurons. Would it be possible to perform retrograde staining of lumbar ascending propriospinal neurons and show that these cells express 5-HT7 receptors but not 5-HT2 neurons?

We agree with the Reviewer that motoneuron identity was not supported by our results. Thus we took care to modify the results description and figure legend regarding this issue by pointing that in the ventral horn there we identified immunopositive big cells that presented motoneuron phenotype.